# Dietary Fluoride Intake by Children: When to Use a Fluoride Toothpaste?

**DOI:** 10.3390/ijerph18115791

**Published:** 2021-05-28

**Authors:** Adriano Casaglia, Maria Antonietta Cassini, Roberta Condò, Flavia Iaculli, Loredana Cerroni

**Affiliations:** 1Department of Clinical Science and Translational Medicine, University of Rome Tor Vergata, Via Montpellier 1, 00133 Rome, Italy; adriano.casaglia@gmail.com (A.C.); roberta.condo@uniroma2.it (R.C.); cerroni@uniroma2.it (L.C.); 2Department of Oral and Maxillofacial Science, Sapienza University of Rome, Via Caserta 6, 00161 Rome, Italy; flavia.iaculli@uniroma1.it; 3Department of Neuroscience and Reproductive and Odontostomatological Sciences, University of Naples “Federico II”, Via Pansini 5, 80131 Naples, Italy

**Keywords:** fluoride intake, dental fluorosis in children, drinking water, fluoride toothpaste for children, fluoride in food

## Abstract

Fluoride is recommended for its cariostatic effect, but excessive fluoride intake may have health risks. Increased prevalence of dental fluorosis in areas with low fluoride content in drinking water has been attributed to the inappropriate excessive intake of fluoride supplements (tablets and drops) and toothpaste ingestion. The aim of the present study was to estimate the fluoride intake and the risk of fluorosis in children (6 months–6 years) in the Castelli Romani area (province of Rome, Italy), which is volcanic, therefore with a higher concentration of fluorine. Measurements of the fluoride content in drinking water, mineral waters, vegetables and commercial toothpaste for children were performed. The fluoride concentrations of all samples were determined using a Fluoride Ion Selective Electrode (GLP 22, Crison, Esp). Data were analyzed by descriptive statistics. Differences between samples were determined by Student’s *t*-test. The fluoride content in tap water samples collected from public sources averaged from 0.35 to 1.11 ppm. The Pavona area showed the highest content of fluoride with respect to the others (*p* ≤ 0.05). The fluoride content in mineral water samples averaged from 0.07 to 1.50 ppm. The fluoride content of some vegetables showed increased mean values when compared to control vegetables (*p* ≤ 0.05). Within the limitations of the present study, considerations should be made when prescribing fluoride toothpaste for infants (6 months–4 years) in the areas with high fluoride content, because involuntary ingestion is consistent.

## 1. Introduction

The cariostatic effect of fluoride has been studied since the 1930s [1]. The World Health Organization (WHO) approved fluoride as a preventive measure against dental caries in 1969 [2] and recommended its appropriate use in 1994 and 2010 [3,4]. This has also led to increased development of fluoride-releasing materials as a strategy in the prevention or inhibition of caries development and progression, in addition to the well-established curing and finishing procedures aimed at reducing biofilm formation [5,6,7]. The standards for daily fluoride intake in infancy and early childhood are determined using indices indicating whether there has been an increase in tooth resistance against caries. The resistance is due to pre-eruptive maturation of the enamel of permanent teeth and improvement in its crystalline properties. Excess fluoride intake from drinking water and various dental hygiene products have potential health risks [8,9,10]. In fact, prolonged excessive intake of fluoride has been associated with fluorosis [11]. Fluorosis is a degenerative and progressive disorder that adversely affects many other organs, such as the bone biomechanical properties, thyroid, kidney, liver, lung and brain [12,13,14,15,16,17]. Numerous studies have been conducted on the effect of excess fluoride on neurological development. Most of these investigations, which support the neurotoxic effects of fluoride, have been performed on animals and demonstrate the generation of free radicals and alterations in the level of neurotransmitters in the brain [13,18,19]. These changes may interfere with the normal development of the Central Nervous System (CNS) during fetal and early childhood development [20]. Based on these considerations, Recommended Dietary Allowances (RDAs) [21] have been issued in the United States, and tolerable intakes of major nutritional elements, minerals, and trace elements are discussed therein and regulated. Fluoride intake should not substantially exceed the tolerable Upper intake Level (UL) of the appropriate range [21], and an evaluation to establish new criteria [22,23] for Dietary References Intakes was requested. The optimum fluoride intake in children was initially estimated from 0.05 to 0.07 mg/kg of a child’s weight/day; this range has been revised repeatedly with regard to the possible total amount of fluoride intake from food and swallowed toothpaste [24,25,26,27,28]. This amount has been determined by studies that have shown that opacities of the permanent anterior teeth might develop, even when increased intake of fluoride occurs in children over 3 years [29]. Increased prevalence of stains on permanent teeth have been observed in the U.S.A. and Australia, including areas with low fluoride content in drinking water, and has been attributed to the inappropriate intake of fluoride supplements (tablets, drops). This increased staining has led to the decision that the original “optimal” range of daily intake was set at the upper limit [26]. Therefore, fluoride intake from food is a factor closely related to the use of fluoride for the prevention of dental caries and the estimation of daily fluoride intake in infants from foods is therefore needed for planning a prevention program [30].

The clinical features of dental fluorosis are well-known, but the role of fluoride in the pathogenesis of faulty mineralization and the period of maximum risk is still unclear [31]. Most of the studies in individuals who have been introduced to fluoride at different ages seem to indicate that the 1st year of life is a crucial period for the development of dental fluorosis [30,32]. Exposure to fluoride supplements could be a risk factor and cannot be ignored in the pathogenesis of dental fluorosis of the permanent teeth [33,34]. Weaning in most cases starts at 5–6 months and weaning food is normally prepared by adding tap water to ready-made powder and, if the water is high in fluoride, weaning food may be an early and relevant carrier of fluoride from the water [35]. The use of fluoridated toothpaste before 14 months of age, if fluoride intake from diet is higher than RDA, is the main risk factor associated with mild forms of dental fluorosis in the maxillary central incisors [25,36].

The purpose of this research is to assess the risk of fluorosis in children (6 months–6 years) in the area of Castelli Romani (province of Rome, Italy). The content of fluoride in commercial toothpaste for children, drinking water, some mineral waters and vegetables prepared with different types of water was measured. 

## 2. Materials and Methods

### 2.1. Sample Collection

Tap water samples were collected from public sources in five different residential areas near Rome called Castelli Romani (Albano, Cecchina, Pavona, Marino and its hamlet Cava dei Selci) in 2016. For each area, ten samples of 50 mL were collected from public fountains located in different streets every 10 days for a period of more than three months. All samples were analyzed on the same day of collection.

Ten commercially available brands of Italian bottled mineral water were selected and considered for fluoride content evaluation. The mineral waters were Claudia, San Pellegrino, San Benedetto, Sangemini, Levissima, Vera, Uliveto, Panna, Ferrarelle and Egeria. All samples were stored sealed in their original containers until the fluoride analysis was performed. The fluoride content of each sample was determined twice.

Five types of vegetables, such as eggplant, tomato, pepper, lettuce and chard were chosen from the same areas where the tap waters were collected. The vegetables of control samples were obtained from a low fluoride area and domestic tap water from cultivation areas was taken and used to prepare the vegetables.

Ten commercially available toothpastes for children in Italy were selected and considered for fluoride content evaluation (Table 1). All toothpastes contained silica and/or hydrated silica as the abrasive. All samples were number coded so that the investigators were blind to the type of water, vegetable or toothpaste analyzed. 

### 2.2. Sample Preparation of Tap Water and Mineral Water

After shaking the sample container, 10 mL of tap water was taken and mixed with 1 mL of Total Ionic Strength Adjusting Buffer III (TISAB III, Crison, Esp). The same amount (1 mL) of conditioner TISAB III was also added to samples of mineral water (10 mL). 

### 2.3. Vegetable Sample Preparation

Five types of vegetables were chosen from the same areas where the tap water was collected, and each vegetable was weighed (50 g) and boiled for 30 min in 250 mL of tap water coming from the same area, and then blended. The samples obtained were divided into two groups: a first sample of 25 mL was examined as it was, and the second sample of 25 mL had 10% 0.01 M orthophosphoric acid (H_3_PO_4_) added and was left under agitation at 4 °C for 24 h. The obtained homogenates were centrifuged at 2200 rpm for 15 min, and 10 mL of supernatant was conditioned with 10% TISAB III.

### 2.4. Toothpaste Sample Preparation

A quantity of 500 mg of each toothpaste sample was collected and mixed in test tubes (Falcon, 10 mL) with 10 mL of distilled water to obtain a homogeneous solution. For each toothpaste, four different test tubes were created, each one for a different solution treatment: A was suspension after mixing, B was the supernatant after centrifugation and heat-treated, C was the supernatant after centrifugation cold-treated, and D was the residue toothpaste + 2 mL of distilled water. From the test tube containing the homogeneous solution of distilled water and toothpaste, 2 mL of solution was taken to obtain sample A (suspension after mixing). The tubes with the remaining 8 mL of solution were centrifuged for 10 min at 3000 rpm. Following the centrifugation, another 2 mL + 2 mL of supernatant was taken and placed, respectively, in tubes B and C (supernatants heat- and cold-treated). Finally, in the test tube containing the residue, the remaining 4 mL of the solution was taken and placed in 2 mL of distilled water to obtain sample D (residue toothpaste). For samples A, B and D, 2 mL of 2 M hydrochloric acid (HCl) was added, and tubes were incubated for one hour at a temperature of 45 °C. After neutralization with 4 mL of 1 M sodium hydroxide (NaOH), a 10% *v*/*v* solution of conditioning TISAB III was added (0.8 mL). A different protocol was followed for sample C (supernatant, cold-treated), directly adding 4 mL of 1 M NaOH, 0.8 mL of TISAB III and 2 mL of 2 M HCl without any incubation.

Analyses were carried out in duplicate, according to a modified protocol (Pearce 1974) [37].

### 2.5. Measuring Fluoride Concentration

All samples were number coded so that the investigators were blind to the type of water, vegetable or toothpaste analyzed. Fluoride standards ranging from 0.1 to 100 parts per million (ppm) were used to calibrate the measurement. The fluoride concentrations of all samples were determined using a Fluoride Ion Selective Electrode (GLP 22, Crison, Esp) according to these steps: 1. withdrawal of the full content of the conditioned specimen from the tube; 2. mixing at 500 rpm for 1 min with laboratory shaker (MR 2002, Heidolph); 3. measuring fluoride content under conditions of constant temperature and stirring (Mivar ALC) and, at the end of each measurement, the electrodes were rinsed with distilled water and dried.

Data were analyzed by descriptive statistics. The paired *t*-test was employed to compare the different measurements of each sample. Differences between samples were determined by Student’s *t*-test. The level of statistical significance was set at 0.05.

## 3. Results

### 3.1. Tap Water

Tap water samples were collected in 2016 from public sources in five residential areas near Rome called Castelli Romani (Albano, Cecchina, Pavona, Marino and Cava dei Selci) over a period of three months.

Results for fluoride content are shown in Table 2. Albano showed a general media of 0.40 ppm, which was the lower average measurement of fluoride in the area, with the highest and lowest sources, respectively, of 0.47 ppm and 0.35 ppm fluoride. Cecchina, a village of 12,000 inhabitants, had average quantities of fluoride of 0.55 ppm, with peak values of 0.67 ppm F. Pavona showed instead that the highest average measurement of fluoride over the period was 0.97 ppm; it must be emphasized that the P2 source reached average levels of 1.11 ppm F, which was more than fluoridated water according to last year’s standards. Cava dei Selci showed 0.42 ppm F, while Marino showed only slightly higher values of 0.46 ppm, with a peak value of 0.53 ppm F. Pavona showed 142% more fluoride in tap water than Albano (*p* ≤ 0.05).

### 3.2. Bottled Mineral Water

Ten bottled mineral waters were chosen among those most used in Italy, and locally, and analyzed to assess the total quantity of fluoride in 2004, 2005, 2014 and 2016. Results were compared to the only similar published research (56) and to the values declared on the labels (Table 3). Two brands (San Pellegrino and Vera) have not declared the fluoride content on the label over the last 12 years, even if both mineral waters showed a medium fluoride content. Three brands (Levissima, Panna and San Benedetto) began to report the data on the label from 2005, while Egeria started reporting much later. Ferrarelle, Claudia and Uliveto have always reported the data. Sangemini stopped reporting the fluoride content for 10 years after 2005. 

All mineral waters with a medium-high fluoride content exceeded the value on the label and the fluoride levels were not detectable in bottled waters with a low content of fluoride in 2014 and 2016. Among the mineral water with high fluoride content, Egeria differed from the data on the label. In fact, it had an average fluoride content approximately 49% higher than that reported on the label.

Egeria in 2016 is also the water that had a 43% higher mean annual value than the label. The water that showed the highest discrepancy with respect to the label was Levissima with 110% more fluoride in 2004. Four of the analyzed mineral waters (Uliveto, Ferrarelle, Claudia and Egeria) exceeded the concentration limit for fluoridated tap waters (1 ppm) from 2014 to 2016.

The peak values exceeded the mean values by more than 12% and the highest peak value found in our measurements was 1.79 ppm for Egeria.

### 3.3. Vegetables

The fluoride content of the vegetables prepared with different waters coming from different areas are shown in Table 4. The control sample showed a remarkable concentration of fluoride, which can be justified as vegetables which came from a low fluoride area had been boiled in tap water from selected areas of Albano, Cecchina and Pavona. Greater amounts of fluoride in samples of tomato, eggplant and chard chosen from high fluoride areas could be cultivated in a soil rich in fluoride. 

Pepper and lettuce were more stable and did not present with increased fluoride content when compared to the means obtained in the examined area. They showed very limited increases when comparing the peak values (respectively +2% and +12%). Eggplant and chard are very influenced by soil composition. They were higher by 78.6% and 38.8%, respectively, compared to the control values (*p* ≤ 0.05), and 88% and 72%, respectively, if we consider the peak values.

Tomato was the vegetable that acquired more fluoride from the soil and tomatoes grown in Castelli Romani were 88.4% higher than tomatoes grown in low fluoride areas when boiled in the same tap water. Tomatoes from Albano showed a 226% increase in fluoride content.

Finally, when we consider the overall measurements of the five vegetables obtained in each municipality, we found that vegetables from Albano had 16.3% higher fluoride content, vegetables from Cecchina had 25.4% increased content, and vegetables from Pavona had 42.0% increased content compared to control vegetables.

### 3.4. Toothpastes

In Table 1, the most used toothpastes among children in the examined area of Castelli Romani are reported. Among the examined toothpastes, 20% contained fluorides (F) as Sodium Monofluorophosphate (MFP) and 70% as Sodium Fluoride (NaF). In most toothpastes (66.6%), the Total Fluoride (TF) concentration found in the analysis coincided with that declared on the product label (≤5%). Disagreement (≥18%) between the concentration declared and that found was observed for toothpastes C, E and I. The TF found in the F toothpastes varied from 525 ± 29.33 to 860 ± 75.53 ppm F. In 89.9% of F toothpastes, the Total Soluble Fluoride (TSF) was lower than TF, indicating that part of F was insoluble (Table 5). It should be pointed out that measurements of all toothpastes containing 500–600 ppm NaF/MFP confirmed the data reported on the label with good accuracy. In contrast, toothpastes containing NaF at a declared quantity of 1000 ppm had measured quantities that were 20% lower. Excipients or other ingredients contained in the toothpaste composition are probably able to interact with the fluoride over certain concentrations, reducing their presence with intermediate compounds.

The F ion (FF) was detected in all toothpastes except A, B and H, and the F ion concentration found in the analysis coincided with that of TSF (disagreement ≤ 1%). The different content of TF between 1000 ppm or 500 fluoride toothpaste was not significant.

### 3.5. Total Fluoride Intake

Table 6 summarizes the data obtained from “Livelli di Assunzione di Riferimento di Nutrienti ed energia per la popolazione italiana (LARN)” for the “Società Italiana di Nutrizione Umana (SINU)”, which indicates the fluoride RDA for Italian children. It was also possible to extrapolate a standard weekly dietary pattern for children, which allowed us to obtain data from the 4th column for the estimated mean fluoride intake from generic food. The last column reports hypothetic fluoride consumption from generic food with vegetables coming from Castelli Romani as reported in Table 5.

Results from years 0 to 1 are not reported because there were no reliable studies on the amount of fluoride in breast milk in Italy. Breast milk contains a generally low amount of fluoride, but this amount should still be sufficient to bring values inside RDA. If the average water consumed by age group (mL) is multiplied by the content of fluoride detected in tap water from the areas studied in 2016, it is possible to estimate the amount of fluoride ingested by children in those areas (Table 7).

The data from food and tap water from the studied areas were combined, and the results are reported in Table 8. All values exceeded RDA up to 4 years of age. Between 4 and 6 years of age, Pavona exceeded RDA while the other 4 areas were dangerously close to the upper margin of RDA and exceeded it on average.

The results from food and mineral water were also combined in Table 8, which describes a diet in which the only source of water is bottled water. In this case, every mineral bottled water with high, medium-high and medium-low fluoride content led to exceeding the RDA. Only drinking mineral water with low fluoride content made it possible to remain in the correct range, especially between 4 and 6 years of age. 

Table 9 reports the total involuntary ingestion of fluoride from children using different types and amounts of the different classes of toothpastes (500–600–1000 ppm F).

Involuntary ingestion is inversely proportional to the children’s age, which leads to the fact that younger children ingest larger quantities of fluorides.

## 4. Discussion

Excessive fluoride ingestion during tooth formation is the cause of dental fluorosis, which results in hypoplasia or hypomineralization of tooth enamel and dentine. It is clear that the main sources of fluoride associated with increased dental fluorosis are fluoridated supplements, dentifrices, water and processed baby foods consumed before six years of age [38,39]. 

Groundwater chemistry is basically due to the ionic components transported by rain to the karst dissolution of the crossed rocks and to the superficial contaminants present on the ground (such as herbicides, fertilizers, etc.) [40]. Geographical areas with groundwater fluoride contamination are mostly characterized by the presence of crystalline basement rocks and/or volcanic bed-rocks [41].

The chemical composition of the aquifers is extremely variable and depends on several factors: the depth of the aquifer itself, the soil composition, its characteristic permeability or impermeability and the use that humans make of that area. All these variables determine the type of groundwater system. Average rainfall in a given period influences the filling of an aquifer and its chemism in a complex hydrogeological framework. Therefore, the chemical composition of bottled mineral water may depend on the frequency and the amount of rainfall [42].

For example, with the same rainfall falling within a month in a given area, if the storms are few and violent due to the increased surface run, the water table is recharged less than the situation in which there are numerous rainy days of low intensity. The latter can more easily infiltrate the ground.

Furthermore, the temperature of the rain plays a role in the relationship between filtration and evaporation, where the latter increases with increasing temperature, reducing the water that infiltrates the soil.

Finally, the more or less elevated soil aridity at the time of the rain influences permeability in a directly proportional way. Therefore, the net impact in a given location will depend on the variation in total rainfall, the variability of the precipitation events themselves and on the temperature and conditions of the soil in an extremely complex hydrogeological situation [40,41,42].

The fluoride concentration in bottled waters from the United States is insufficient to contribute to caries prevention, fluoride concentrations were generally low (mean, 0.11 ppm) [43]. The fluoride concentration in Italian mineral waters is higher in 80% of brands (>0.3 ppm).

Mineral water is usually chosen more for taste than for the information on the label. Increasing consumer awareness, by spreading the importance of the data on the mineral water label, should be a primary goal.

It is important to remember that the Legislative Decree 25 January 1992, n 105, Art 11.6, indicates that the information provided on the label should be updated at least every 5 years. Italian legislation also requires writing on the label “fluoridated” on bottles of mineral water that contain more than 1.0 ppm of fluoride, and “not suitable for regular consumption by infants and children under 7 years of age” for mineral or spring water that contains more than 1.5 ppm.

These indications are often ignored. In fact, some mineral waters, even though presenting average quantities of fluoride, do not include the data on the label (San Pellegrino and Vera). Many others have reported the value only for some periods. Most likely, the first possible improvement would be to standardize the labels with a model that includes all the most important characterizing elements. This standardization would improve the possibility for the consumer to compare the basic characteristics of bottled mineral water.

Considering the extreme variability in meteorological conditions, which produce important hydrogeological effects on groundwater, a label updated only every 5 years describes the characteristics of mineral water in an absolutely ineffective and imprecise manner. We suggest the creation of a National Water Database with all data from all commercial drinking waters to be updated even more often than every 5 years. Customers could constantly check the values. A QR code on every bottle could point to the up-to-date values. The same database should be “filled” monthly by the Municipalities. Finally, more control should be implemented so that the terms “fluoridated” and “not suitable for regular consumption by infants and children under 7 years of age” are truly reported in the labels.

Little is known about dietary fluoride intake from foods and beverages in Italy, especially in areas with the highest fluoride content in soil and water. This lack of information was the reason for studying a precise area surrounding Rome, called Castelli Romani. Here, the soil is very rich in various mineral components due to its volcanic origin.

The starting point was to hypothesize the risk of ingesting excessive amounts of fluoride for a child who lives in one of the aforementioned areas. The calculation was made assuming that the child eats local foods cooked with tap water and drinks tap water.In fact, the hypothesis of finding large quantities of fluoride in the soil and, consequently, in irrigation wells and the aqueduct, has been confirmed. This hypothesis was suggested, not only by the chemical and geological characteristics of the volcanic territory but also by the recent quantitative assessments carried out by the municipal authorities engaged in the search for the presence of arsenic in the aqueduct. The results confirm that four out of five municipalities present quantities in line with expectations, i.e., a medium-high level of fluoride in drinking water (from 0.4 to 0.55 ppm). There has been significant oscillation in the measurements of fluoride in tap water. It is likely this phenomenon is due to the meteorological conditions of the period, to rainfall, which is finally reflected on the composition of the aquifer.

It should also be noted that the amount of fluoride detected in the water of the wells examined was extremely high. This water is currently used for breeding and irrigation. All this water was not the subject of this research, and further research is suggested.

It is important to understand that in the Municipality of Pavona, the average fluoride value over the three months was as high as the value used in the water with added fluoride in areas where this compound is otherwise low (0.97 ppm). In essence, a child in Pavona drank, at least in those three months, the exact amount as from fluoridated water.

For this reason, it would be a good idea that the municipality or, even better, the company that manages the aqueduct, publishes the results of the periodic chemical analysis allowing the final user to acquire greater knowledge of the “product” water.

Analyzing the data in Table 5, it is clear that some foods contain, on average, more fluorine than others. For example, among the 5 vegetables tested, chard is a type of plant that has a high concentration of fluorine, both in the geographical areas of Castelli Romani, where there is a high presence of this element, and in control areas, which instead have a low concentration of fluorine in water and soil. Naturally, the presence of fluorine in the chard was higher when coming from Castelli Romani compared to the control area but, in any case, chard is characteristic of those crops that absorb and retain more fluorine than others. The tomato always had very low concentrations of fluorine and, in absolute value, is a food poor in fluorine. In relative value, however, it is strongly dependent on the territory in which it is grown. In fact, taking the highest value found in the cultivations of Castelli Romani and the lowest value of the rural control area, it is up to 4 and a half times richer in fluorine.

In summary, the concentration of fluorine found in food is proportional to the quantity present in the territory and is mediated by the intrinsic characteristics of the individual vegetable. This combination of factors constitutes a biological limit, which causes some foods to be strongly influenced by external factors, the environment, pollution or by more or less sustainable agriculture choices.

The fluoride in the water in the wells is proportional to that found in the aqueducts, and it is the level that influences the quantity of fluorine present in the cultivations irrigated with that water. The water in the aqueduct is normally used for cooking dishes and, in many families, also as drinking water. This use will lead to an additional load of fluorine in those areas where the aquifers are richer. By monitoring the area of Castelli Romani with a high concentration of fluoride over the years, we found a territorial reduction between 2010 and 2013 when episodes of dangerous peaks of arsenic in the aquifer were reported. The pollution was resolved by intervening with maintenance works on the aqueduct network and installing appropriate filters. This process could also have affected the presence of other elements, including fluorine. 

When the fluoride results for tap water and vegetables are inserted in a standard diet according to the directives of the LARN, compared to the age of the children, quite clear results are obtained.

Up to 4 years of age, a balanced diet, according to specialized indications, based on the preparation of local foods and the use of tap water as the main source of hydration, is already sufficient to exceed the fluorine RDA in all areas taken into consideration. In areas with high concentrations of fluorine, the use of mineral waters with low fluoride content, as a substitution for tap water, is sufficient to bring the values back into the RDA.

In the age group between 5 and 6 years the situation is attenuated, although these values are still high, probably due to physical growth that includes increased weight. Only the children in Pavona have the same situation as the 0 to 4 age group previously described. In other areas, the values remain high but closer to the suggested maximum RDA limits. In these cases, the low and medium-low levels of fluoride mineral water allow the intake to remain within the RDA. In contrast, mineral waters with a medium-high and high fluoride content, exceeding the average values present in tap water, further increase the quantities of fluorine taken in all age groups. 

Many toothpastes for children show that they can be used from 0–6 years, which is not correct information. Parents are generally not educated about the right amount of toothpaste to be used based on the age of the child, resulting in an excessive amount used. The use of topical fluoride is not recommended in children before the end of the first year of life [44].

This recommendation is not always given in the instructions for toothpastes that were tested. In two cases, the age recommended for use was not clear (BeJ and SCK). Otherwise, a toothpaste containing 1000 ppm of NaF showed 0–6 years as the recommended age range (CPC).

It would be useful for pediatricians to know this information, including that concerning the involuntary ingestion of toothpaste based on age to transmit information regarding oral hygiene and the correct use of related products to children’s parents.

Water for human use may come from natural or artificial basins, streams or rivers. Bottled mineral waters instead came almost exclusively from natural sources, such as aquifers.

According to the above-mentioned recommendations for infants (6 months–1 year) in the examined areas, in which involuntary ingestion increases, fluoride toothpastes should be avoided. Uptake of fluoride during the pre-eruptive stage of enamel formation is dentally important owing to the increased risk of dental fluorosis development if systemic ingestion of fluoride is chronically excessive in early infancy. Assessment of TF exposure in young children has been recommended [35,45]. Children 1–4 years of age probably will not receive great benefits from the presence of fluoride in toothpaste formulations. Children between 5 and 6 years of age progressively reduce involuntary ingestion of toothpaste, and therefore of fluoride, and ingest less fluoride through food. For this age group, the use of fluoride-containing toothpaste could be acceptable as long as they use one with reduced content (500–600 ppm) and pay attention to the quantity administered (pea size or smear) to reduce involuntary ingestion as much as possible. Because there are many sources of fluoride in the water supply and in processed food, it is essential that all potential sources of fluoride be assessed before prescribing a toothpaste containing fluoride [46].

A warning shall also be placed on the toothpaste tubes, asking the customers to check their tap water’s fluoride values before using the product.

## 5. Conclusions

The results obtained in this study allow us to draw the following conclusions:

Tap water—The Municipalities should periodically update the data concerning the content of chemical and mineral substances present in the water (including fluorine);

Mineral water—The time required to update data on the label should be reduced to a maximum of 2–3 years and more complete data on the label should be introduced that contains not only a single measurement for each item but the average of periodic surveys taken over a larger reference period;

Vegetables—The concentration of fluorine found in food is proportional to the quantity present in the territory but is mediated by the intrinsic characteristic of the individual vegetable. This relation constitutes a biological limit, which causes some foods to be strongly influenced by external factors, the environment, pollution or by more or less sustainable agriculture choices. The presence of fluoride in well water is proportional to that found in the aqueducts; 

Fluoride supplements—We can finally conclude that the indiscriminate prescription of fluoride supplements by family doctors and pediatricians for children of any age group should be avoided. This restriction is valid in general and even more in areas with similar territorial characteristics to those examined by this study unless a preliminary investigation is carried out regarding the specific eating habits of the child; 

Toothpastes—We can conclude that they are to be avoided at least up to 1 year of age. Between 1 and 4 years they can begin to be introduced but making sure they do not contain fluoride. The daily requirement is covered through nutrition. Attention must be even more rigorous in hydrogeological areas rich in this element. From age 5 and up, the risk of accidental toothpaste ingestion decreases sharply, and the danger of fluorosis will be more limited. 

However, it is always necessary to ascertain the concentration of fluoride present in the water (bottled or tap) used for the preparation of food and given to drink to children. GPs, pediatricians and parents should be made more aware of the presence of fluoride in their geographical area of residence, in the mineral waters used and in children’s toothpastes.

## Figures and Tables

**Table 1 ijerph-18-05791-t001:** Reference table for toothpaste, declared fluoride content and information about the formulation (Nd = Not declared).

Code	Toothpaste	Recommended Age	Type of Fluoride	Declared Fluoride (ppm)
A	BC	+1	MFP	500
B	BJ	0–13	Nd	0
C	CPC	0–6	NaF	1000
D	API	3–5	NaF	500
E	CS	2–6	NaF	1000
F	BeJ	Nd	NaF	500
G	GK	2–6	NaF	500
H	SCK	Nd	MFP	600
I	MK	+3	NaF	1000
L	OBS	2–4	NaF	500

**Table 2 ijerph-18-05791-t002:** Fluoride content (ppm) in tap water from public tap standpipes in 2016. Related areas of Castelli Romani: A—Albano, C—Cecchina, P—Pavona, S—Cava dei Selci, M—Marino. Each number indicates different sources.

Public Sources	2016
A1	0.37 ± 0.03
A2	0.43 ± 0.10
A3	0.47 ± 0.12
A4	0.35 ± 0.12
A5	0.38 ±0.05
C1	0.59 ± 0.08
C2	0.67 ± 0.14
C3	0.40 ± 0.06
P1	0.72 ± 0.12
P2	1.11 ± 0.03
P3	1.07 ± 0.05
S1	0.37 ± 0.05
S2	0.47 ± 0.06
M1	0.38 ± 0.06
M2	0.53 ± 0.07

**Table 3 ijerph-18-05791-t003:** Fluoride content in bottled mineral water available in Italy. The results from Itro et al. 2000, labels in 2004, 2005 and 2016, and experimental measurements in 2004, 2005, 2014 and 2016 (Nd: Not declared).

Mineral Water	Results Itro A. 2000	Label 2004	Label 2005	Label 2017	Results 2004	Results 2005	Results 2014	Results 2016
Ferrarelle	0.70	1.00	1.00	1.10	0.98	1.24	1.37	1.30
Egeria	1.60	Nd	Nd	1.05	1.00	0.76	1.67	1.50
Sangemini	0.26	0.20	<0.20	0.29	0.32	0.30	0.32	0.30
San Pellegrino	0.70	Nd	Nd	Nd	0.52	0.20	0.57	0.50
Panna	0.98	Nd	<0.10	<0.10	0.16	0.05	0.07	0.10
Levissima	0.30	Nd	0.20	0.20	0.42	0.06	0.28	0.30
Claudia	2.10	1.30	1.30	1.45	0.84	0.35	1.49	1.50
San Benedetto	<0.10	Nd	0.06	<0.10	0.09	0.07	0.09	0.07
Uliveto	1.40	1.00	1.00	1.00	1.30	0.80	1.03	1.14
Vera	0	Nd	Nd	Nd	0.46	0.05	0.14	0.25

**Table 4 ijerph-18-05791-t004:** Fluoride content in vegetables (μgr/gr = ppm) from areas of Castelli Romani: A (Albano), C (Cecchina), P (Pavona), CTR (crops in low fluoride areas).

Vegetables	A	C	P	Mean Value	CTR
Pepper	2.47 ± 0.19	3.42 ± 0.19	3.10 ± 0.16	3.00 ± 0.18	3.04 ± 0.16
Tomato	3.10 ± 0.28	0.76 ± 0.19	1.52 ± 0.19	1.79 ± 0.22	0.95 ± 0.19
Eggplant	5.13 ± 0.47	6.08 ± 0.47	6.10 ± 0.38	5.77 ± 0.44	3.23 ± 0.38
Lettuce	5.41 ± 0.37	5.20 ± 0.32	6.05 ± 0.19	5.88 ± 0.29	4.91 ± 0.37
Chard	6.46 ± 0.19	8.87 ± 0.55	10.78 ± 0.55	8.70 ± 0.43	6.27 ± 0.57

**Table 5 ijerph-18-05791-t005:** Fluoride concentration of different toothpastes, expressed as ppm (parts per million). TF (total fluoride), TSF (total soluble fluoride), FF (free fluoride) and IF (ionic fluoride).

Toothpaste	TF	TSF	FF	IF
A	550 ± 7.18	400 ± 57.02	0	0
B	0	0	0	0
C	760 ± 75.53	655 ± 62.84	765 ± 73.80	0
D	540 ± 2.76	430 ± 27.91	540 ± 42.60	2 ± 0.24
E	825 ± 17.93	825 ± 11.51	833 ± 15.30	3.25 ± 1.49
F	530 ± 29.7	470 ± 46.35	565 ± 20.49	0
G	525 ± 20.18	500 ± 40.12	510 ± 36.46	0.52 ± 0.50
H	640 ± 17.18	625 ± 16.89	0	0.63 ± 0.55
I	825 ± 10.28	785 ± 18.01	820 ± 44.06	20.80 ± 0.94
L	525 ± 29.33	465 ± 37.42	515 ± 35.49	6.1 ± 1.63

**Table 6 ijerph-18-05791-t006:** Summary table for food obtained from “Livelli di Assunzione di Riferimento di Nutrienti ed energia per la popolazione italiana (LARN)”, Società Italiana di Nutrizione Umana (SINU) for Italian children.

Age	Medium Weight	RDA F (mcg/Day)	F (mcg/Day) from Food	F (mcg/Day) from Food Corrected from Table 5
0–0.5	3–8	100–500	Not reported	Not applicable
0.5–1	6–10	200–900	166–254	Not applicable
1–2	8–15	500–1000	558–821	672–988
2–4	10–20	750–1100	615–984	740–1185
4–6	14–25	1100–2000	784–1176	944–1416

**Table 7 ijerph-18-05791-t007:** Mean calculated daily fluoride ingestion (expressed as mcg/Die) obtained from Table 3.

AGE	Daily Water Intake (mL) LARN	From Water in Albano mcg/Die	From Water in Cecchina mcg/Die	From Water in Pavona mcg/Die	From Water in Cava De Selci mcg/Die	From Water in Marino mcg/Die
1–2	1200–1500	480–600	660–825	1164–1455	504–630	552–690
2–4	1380–1980	552–792	759–1089	1338–1921	580–832	635–911
4–6	1800–2300	720–920	990–1265	1746–2231	756–966	828–1058

**Table 8 ijerph-18-05791-t008:** Dietary intake of fluoride in studied areas (food plus water and food plus mineral bottled water); mineral waters were chosen as representatives of high (Egeria-Claudia), medium-high (Uliveto), medium-low (San Pellegrino) and low (San Benedetto) in fluoride content.

	Fluoride (mcg/Die)									
	Food Plus Tap Water						Food Plus Mineral Bottled Water			
AGE	RDA F (mcg/die)	Albano	Cecchina	Pavona	Cava de Selci	Marino	Egeria/Claudia	Uliveto	Pellegrino	San Benedetto
1–2	500–1000	1152–1588	1332–1813	1836–2443	1176–1618	1224–1678	2472–3238	2040–2698	1272–1738	756–1093
2–4	750–1100	1292–1977	1499–2274	2078–3106	1320–2017	1375–2096	2810–4155	2313–3442	1430–2175	837–1324
4–6	1100–2000	1664–2336	1934–2681	2690–3647	1700–2382	1772–2474	3644–4866	2996–4038	1844–2566	1070–1577

**Table 9 ijerph-18-05791-t009:** Amount of toothpaste and fluoride involuntarily ingested from children using smear, pea-size or the amount reported in the literature from two brushings a day. * Calculated on media for total fluoride measured (534 ppm) in toothpastes with declared fluoride of 500 ppm; ** Calculated on media for total fluoride measured (640 ppm) in toothpastes with declared fluoride of 600 ppm; *** Calculated on media for total fluoride measured (837 ppm) in toothpastes with declared fluoride of 1000 ppm.

		Involuntary Ingestion of Toothpaste Every Brushing (gr/Day)				Fluoride Ingestion (mcg/Day)		
AGE	Ingestion (%)	Smear (0.125 gr)	Pea Size (0.25gr)	Literature (0.35–0.49 gr)	TOTAL	Toothpaste 500 ppm *	Toothpaste 600 ppm **	Toothpaste 1000 ppm ***
0.5–1	60	0.150	0.300	0.420–0.588	0.150–0.588	80–314	96–376	125–492
1–2	50	0.125	0.250	0.350–0.490	0.125–0.490	67–262	80–314	105–410
2–4	46	0.115	0.230	0.322–0.451	0.115–0.451	61–241	74–289	96–378
4–6	34	0.085	0.170	0.238–0.333	0.085–0.333	45–178	54–213	71–279

## Data Availability

Raw data are available on request addressed to corresponding author on reasonable request.

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
