# Peer review of "Dietary Fluoride Intake by Children: When to Use a Fluoride Toothpaste?"

_ijerph, 2021, doi:10.3390/ijerph18115791_

Round 1

Reviewer 1 Report

Dear authors, please find the attached comments for your kind consideration. 

Author Response

REVIEWER 1 

The topic studied is of importance within the field of public dental health, especially in pediatric preventive dentistry. Please find the following comments for your kind consideration. 

- P 1 L 17-28: Abstract is written in general. Please include study design and what kind of statistical analysis were used. 

The Authors are thankful for the comment on the paper. The abstract has been improved according to the Reviewer’s suggestion.

- P 1 L 19: Include the word “excessive” in between “inappropriate intake” to make clear for readers: “inappropriate excessive intake”. 

The suggested edit has been undertaken (page 1, line 22).

- P 1 L 22-23: Please rearrange order of the samples in the abstract, according to your sample collection order in the main text as moving “commercial toothpaste for children” after “vegetables”. 

The suggested change has been undertaken (page 1, line 26).

- P 1 L 44-45: Write in capital letters the “central nervous system”.

The suggested change has been provided (page 2, lines 67-68).

- P 2 L 49-45 and P 4 L 153, and L 157: Write in capital letters the “upper level” and “total 

fluoride”, and “total soluble fluoride”.

All the recommended revisions have been provided.

- P2 L 83, P3 L105, P 3 L109, P 3 L117, P 4 L 138: All starts with capital letters.

All the suggested changes have been undertaken.

- P3 L 110: “five types” and L 118 “toothpastes”.

The suggested edits have been provided.

- P3 Table 1 and P 6 Table 4: In the headnotes of the tables please use the same style of writing 

the headnotes for “(Nd=not declared)/ (Nd: Not declared). 

Headnotes have been provided in the same format as requested. 

- P9 Table 9: In 4th column replace coma by dot (0,300 to 0.300).

The suggested change has been provided within Table 9.

- P9 L 295-306: Discussion part about toothpaste, please move to the end and start discussion on 

water, according to the flow and order of the results. 

Discussion section has been amended as recommended. 

- P 12 L449-451: please rearrange “periodically make public the data” in the sentence. Sounds 

grammatically incorrect. 

The sentence has been edited and provided in a clearer fashion. 

- In the Materials and Methods section: Please include information whether the study conducted 

within a project framework, duration; about conflict of interests due the involvement of commercial 

bottled water, toothpaste companies, presence of IRB etc. 

The present study was not part of a project framework. The conflict of interest has been inserted in the Materials and Methods section as requested; moreover, the presence of the IRB was not necessary, since the present study did not directly involve human subjects.

- What was the inclusion criteria for choosing the 5 types of vegetables as the sample. Why not other vegetables but present 5?

The vegetables were chosen on the basis of their seasonality and because they were locally grown in the areas under study.

Thank you in advance for your time.

Kind regards

Dr. Maria Antonietta Cassini

Department of Clinical Science and Translational Medicine 

University of Rome Tor Vergata 

Via Montpellier 1 - 00133 - Rome -  Italy

Tel: +39 349 7924926

Reviewer 2 Report

The subject of this article is very important and very controversial. There is a lot of discussion in the decision on guidelines for fluoride use and worldwide fluoride continues to be used irrationally.
The article is well written and the design of the study is very interesting,  involving the evaluation of aspects of people's daily lives.
The results are well presented and explored.
The bibliography could be slightly more updated.
The article should be published, without changes, as it leaves an important message on record, and a contribution to the carrying out of more extensive, multicenter studies involving other regions.

Congratulations on the study design
No more comments

Author Response

REVIEWER 2 

The subject of this article is very important and very controversial. There is a lot of discussion in the decision on guidelines for fluoride use and worldwide fluoride continues to be used irrationally.

The article is well written and the design of the study is very interesting, involving the evaluation of aspects of people's daily lives.

The results are well presented and explored.

The bibliography could be slightly more updated.

The article should be published, without changes, as it leaves an important message on record, and a contribution to the carrying out of more extensive, multicenter studies involving other regions.

Congratulations on the study design

No more comments

The Authors are very grateful for the invaluable Reviewer’s considerations. The bibliography has been integrated and updated as suggested.

Thank you in advance for your time.

Kind regards

Dr. Maria Antonietta Cassini

Department of Clinical Science and Translational Medicine 

University of Rome Tor Vergata 

Via Montpellier 1 - 00133 - Rome -  Italy

Tel: +39 349 7924926

Reviewer 3 Report

Dear authors,
This is a nicely carried work of the fluoride levels to which children are subjected. However, I am not sure of the contribution that the present paper can have to the scientific literature.
The introduction section has very high plagiarism rates so should not be evaluated.
The different tables are difficult to understand (table 8), as well as the numbering of the bibliography appears along with years of age, which makes it difficult to read (LINE 32). Why is the full name of the toothpaste not listed? They should be improved
The methodology followed seems correct and following objective criteria.
Conclusion: "For infants (6 months - 4 years) in the examined areas, in which involuntary 26 ingestion is consistent, fluoride toothpaste should be avoided" I believe that this statement should be taken with caution, since many other factors must be taken into account, in addition to the limitations of the present study

Author Response

REVIEWER 3 

Dear authors,

This is a nicely carried work of the fluoride levels to which children are subjected. However, I am not sure of the contribution that the present paper can have to the scientific literature.

The introduction section has very high plagiarism rates so should not be evaluated.

The Authors are very thankful for the Reviewer’s comments and for the appropriate suggested notes. The introduction section has been checked and modified as requested.

The different tables are difficult to understand (table 8), as well as the numbering of the bibliography appears along with years of age, which makes it difficult to read (LINE 32). 

Table 8 has been wholly amended and provided in a clearer fashion as suggested, in order to ease its understanding. Furthermore, the wrong numbering of the bibliography, due to a formatting issue, has been edited.

Why is the full name of the toothpaste not listed? They should be improved

The full name of the toothpastes has not been published due to several reasons. First of all, because the high frequency with which companies modify, even minimally, the formulations of toothpastes mean that the commercial names change very frequently; then, because, regardless of the manufacturing companies taken into consideration in the present study (appropriately chosen between the most widespread), the Authors were interested in investigating the aspect linked to the different fluorine concentrations. This is often repeated by countless companies and ensures that the conclusions of this study remain valid also in the future.

The methodology followed seems correct and following objective criteria.

Thank You for the comment.

Conclusion: "For infants (6 months - 4 years) in the examined areas, in which involuntary 26 ingestion is consistent, fluoride toothpaste should be avoided" I believe that this statement should be taken with caution, since many other factors must be taken into account, in addition to the limitations of the present study.

The sentence regarding the involuntary ingestion of fluoride by infants between 6 months and 4 years has been revised, stressing that many other factors need to be taken into account.

Thank you in advance for your time.

Kind regards

Dr. Maria Antonietta Cassini

Department of Clinical Science and Translational Medicine 

University of Rome Tor Vergata 

Via Montpellier 1 - 00133 - Rome -  Italy

Tel: +39 349 7924926

Reviewer 4 Report

Dear authors, although the article is really very interesting  I suggest some minor corrections to make it more complete before the publication.

Line 32.

The authors shall consider to add some aspects related to Fluoride and restorative procedures. I suggest a sentence like this (I also suggested 3 references to support this sentence)

“This has also lead to an increased developement of fluoride-releasing materials as a strategy in the prevention or inhibition of caries development and progression in addition to the well established curing and finishing procedures aimed at reducing biofilm formation [1,2,3].

  1. Wiegand A, Buchalla W, Attin T. Review on fluoride-releasing restorative materials--fluoride release and uptake characteristics, antibacterial activity and influence on caries formation. Dent Mater. 2007 Mar;23(3):343-62. doi: 10.1016/j.dental.2006.01.022. Epub 2006 Apr 17. PMID: 16616773.
  2. Ionescu AC, Cazzaniga G, Ottobelli M, Ferracane JL, Paolone G, Brambilla E. In vitro biofilm formation on resin-based composites cured under different surface conditions. J Dent. 2018 Oct;77:78-86. doi: 10.1016/j.jdent.2018.07.012. Epub 2018 Jul 17. PMID: 30030124.
  3. Cazzaniga G, Ottobelli M, Ionescu AC, Paolone G, Gherlone E, Ferracane JL, Brambilla E. In vitro biofilm formation on resin-based composites after different finishing and polishing procedures. J Dent. 2017 Dec;67:43-52. doi: 10.1016/j.jdent.2017.07.012. Epub 2017 Jul 24. PMID: 28750776.

Line 217

I would suggest this change:

The control sample showed a remarkable concentration of fluoride, which can be justified while vegetables, which came from a low fluoride area,

Line 336

the word “more” should be removed.

The fluoride concentration in Italian mineral waters is more higher in 80% of brand (> 0.3  ppm).

Line 356

Maybe, thanks to the actual development of technology, the authors could suggest the creation of

a National Water Database with all data from all comemrcial drinking waters to be updated even more often than every 3 years. Customers could check constantly the values.

a Qr code on every bottle could point to the up-to-date values.

Same database should be “filled” monthly byt the Municipalities.

Line 368-370

The starting point was that a child who lives in one of the above-mentioned areas, if he eats local food cooked with tap water and drinks tap water, shows a highly probable risk of taking in excessive amounts of fluoride.

english should be improved in:

The starting point was that a child who lives in one of the above-mentioned areas, could risk an  excessive amounts of fluoride while eating local food cooked with tap water and drinking tap water.

Line 445

The authors could also outline that a warning shall also be placed on the toothpaste tubes, asking the customers to check tap water’s fluoride values before using the product.

Author Response

REVIEWER 4 

Dear authors, although the article is really very interesting I suggest some minor corrections to make it more complete before the publication.

The Authors are grateful for the Reviewer’s considerations. 

- Line 32. The authors shall consider to add some aspects related to Fluoride and restorative procedures. I suggest a sentence like this (I also suggested 3 references to support this sentence). “This has also lead to an increased developement of fluoride-releasing materials as a strategy in the prevention or inhibition of caries development and progression in addition to the well established curing and finishing procedures aimed at reducing biofilm formation [1,2,3].

  1. Wiegand A, Buchalla W, Attin T. Review on fluoride-releasing restorative materials--fluoride release and uptake characteristics, antibacterial activity and influence on caries formation. Dent Mater. 2007 Mar;23(3):343-62. doi: 10.1016/j.dental.2006.01.022. Epub 2006 Apr 17. PMID: 16616773.
  2. Ionescu AC, Cazzaniga G, Ottobelli M, Ferracane JL, Paolone G, Brambilla E. In vitro biofilm formation on resin-based composites cured under different surface conditions. J Dent. 2018 Oct;77:78-86. doi: 10.1016/j.jdent.2018.07.012. Epub 2018 Jul 17. PMID: 30030124.
  3. Cazzaniga G, Ottobelli M, Ionescu AC, Paolone G, Gherlone E, Ferracane JL, Brambilla E. In vitro biofilm formation on resin-based composites after different finishing and polishing procedures. J Dent. 2017 Dec;67:43-52. doi: 10.1016/j.jdent.2017.07.012. Epub 2017 Jul 24. PMID: 28750776.

The Authors agree with the Reviewer’s concern about the missing of a very important aspect about clinical dental practice and prevention. Therefore, the suggested sentence has been provided within Introduction section (lines 39-42) and recommended literature references have been quoted. 

- Line 217. I would suggest this change: The control sample showed a remarkable concentration of fluoride, which can be justified while vegetables, which came from a low fluoride area,

The suggested change has been provided (page 6, lines 291-293).

- Line 336. the word “more” should be removed. The fluoride concentration in Italian mineral waters is more higher in 80% of brand (> 0.3 ppm).

The sentence has been amended as requested (Page 10, line 634).

- Line 356. Maybe, thanks to the actual development of technology, the authors could suggest the creation of a National Water Database with all data from all commercial drinking waters to be updated even more often than every 3 years. Customers could check constantly the values. A Qr code on every bottle could point to the up-to-date values. Same database should be “filled” monthly by the Municipalities.

The suggested aspect has been provided within the Discussion section (page 10, lines 653-656).

- Line 368-370. The starting point was that a child who lives in one of the above-mentioned areas, if he eats local food cooked with tap water and drinks tap water, shows a highly probable risk of taking in excessive amounts of fluoride. English should be improved in: The starting point was that a child who lives in one of the above-mentioned areas, could risk an excessive amounts of fluoride while eating local food cooked with tap water and drinking tap water.

The sentence has been amended as suggested (page 10, lines 663-665).

- Line 445. The authors could also outline that a warning shall also be placed on the toothpaste tubes, asking the customers to check tap water’s fluoride values before using the product.

This important aspect has been provided at the end of the Discussion section (page 12, lines 780-781), as recommended.

Thank you in advance for your time.

Kind regards

Dr. Maria Antonietta Cassini

Department of Clinical Science and Translational Medicine 

University of Rome Tor Vergata 

Via Montpellier 1 - 00133 - Rome -  Italy

Tel: +39 349 7924926

Round 2

Reviewer 3 Report

Work has improved dramatically. I recommend accepting it, checking small writing errors